# Estimated health benefits, costs, and cost-effectiveness of eliminating industrial *trans*-fatty acids in Australia: A modelling study

Matti Marklund[1,2,3]*, Miaobing Zheng[4], J. Lennert Veerman[5], Jason H. Y. Wu[1]

1 The George Institute for Global Health and the Faculty of Medicine, University of New South Wales, Sydney, Australia, 2 Friedman School of Nutrition Science and Policy, Tufts University, Boston, Massachusetts, United States of America, 3 Department of Epidemiology, Johns Hopkins Bloomberg School of Public Health, Baltimore, Maryland, United States of America, 4 Institute for Physical Activity and Nutrition, School of Exercise and Nutrition Science, Deakin University, Geelong, Australia, 5 School of Medicine, Griffith University, Gold Coast, Australia

* mmarklund@georgeinstitute.org.au

## Abstract

### Background

*trans*-fatty acids (TFAs) are a well-known risk factor of ischemic heart disease (IHD). In Australia, the highest TFA intake is concentrated to the most socioeconomically disadvantaged groups. Elimination of industrial TFA (iTFA) from the Australian food supply could result in reduced IHD mortality and morbidity while improving health equity. However, such legislation could lead to additional costs for both government and food industry. Thus, we assessed the potential cost-effectiveness, health gains, and effects on health equality of an iTFA ban from the Australian food supply.

### Methods and findings

Markov cohort models were used to estimate the impact on IHD burden and health equity, as well as the cost-effectiveness of a national ban of iTFA in Australia. Intake of TFA was assessed using the 2011–2012 Australian National Nutrition and Physical Activity Survey. The IHD burden attributable to TFA was calculated by comparing the current level of TFA intake to a counterfactual setting where consumption was lowered to a theoretical minimum distribution with a mean of 0.5% energy per day (corresponding to TFA intake only from non-industrial sources, e.g., dairy foods). Policy costs, avoided IHD events and deaths, health-adjusted life years (HALYs) gained, and changes in IHD-related healthcare costs saved were estimated over 10 years and lifetime of the adult Australian population. Cost-effectiveness was assessed by calculation of incremental cost-effectiveness ratios (ICERs) using net policy cost and HALYs gained. Health benefits and healthcare cost changes were also assessed in subgroups based on socioeconomic status, defined by Socio-Economic Indexes for Areas (SEIFA) quintile, and remoteness. Compared to a base case of no ban and current TFA intakes, elimination of iTFA was estimated to prevent 2,294 (95% uncertainty interval [UI]: 1,765; 2,851) IHD deaths and 9,931 (95% UI: 8,429; 11,532) IHD events

**Funding:** MM and JHYW, are researchers within a National Health and Medical Research Council Centre for Research Excellence in reducing salt intake using food policy interventions (APP1117300). JHYW is supported by a UNSW Scientia Fellowship. MZ is supported by National Health and Medical Research Council Early Career Research Fellowship (GNT1124283). The funders had no role in study design, data collection and analysis, decision to publish, or preparation of the manuscript.

**Competing interests:** The authors have declared that no competing interests exist.

**Abbreviations:** AIHW, Australian Institute of Health and Welfare; AUD, Australian dollar; CHEERS, Consolidated Health Economic Evaluation Reporting Standards; DCIS, Disease Costs and Impact Study; HALYs, health-adjusted life years; ICER, incremental cost-effectiveness ratio; IHD, ischemic heart disease; iTFA, industrial *trans*-fatty acid; NNPAS, National Nutrition and Physical Activity Survey; PIF, potential impact fraction; SEIFA, Socio-Economic Indexes for Areas; TFAs, *trans*-fatty acids; UI, uncertainty interval; WHO, World Health Organization.

over the first 10 years. The greatest health benefits were accrued to the most socioeconomically disadvantaged quintiles and among Australians living outside of major cities. The intervention was estimated to be cost saving (net cost <0 AUD) or cost-effective (i.e., ICER < AUD 169,361/HALY) regardless of the time horizon, with ICERs of 1,073 (95% UI: dominant; 3,503) and 1,956 (95% UI: 1,010; 2,750) AUD/HALY over 10 years and lifetime, respectively. Findings were robust across several sensitivity analyses. Key limitations of the study include the lack of recent data of TFA intake and the small sample sizes used to estimate intakes in subgroups. As with all simulation models, our study does not prove that a ban of iTFA will prevent IHD, rather, it provides the best quantitative estimates and corresponding uncertainty of a potential effect in the absence of stronger direct evidence.

## Conclusions

Our model estimates that a ban of iTFAs could avert substantial numbers of IHD events and deaths in Australia and would likely be a highly cost-effective strategy to reduce social–economic and urban–rural inequalities in health. These findings suggest that elimination of iTFA can cost-effectively improve health and health equality even in countries with low iTFA intake.

## Author summary

### Why was this study done?

- Intake of *trans*-fatty acids (TFAs) is a well-known risk factor of ischemic heart disease (IHD), and several countries have implemented strategies with mandatory limits of TFAs in foods.

- In Australia, intake of TFAs is on average low but does not appear to have decreased in the last decades and is high in certain groups.

- A legislative ban to eliminate industrial TFAs (iTFAs) from the Australian food source is a plausible strategy to reduce disease burden and health inequality.

### What did the researchers do and find?

- We modelled the estimated health effects, costs, cost-effectiveness, and impact on health equality of a nationwide ban of iTFAs in Australia.

- Over the lifetime of the adult population, around 40,000 IHD deaths could be prevented, and about 100,000 health-adjusted life years could be gained.

- The ban was estimated to be cost saving (net cost <0 AUD) or highly cost-effective and could reduce health inequalities in the first 10 years after implementation.

## What do these findings mean?

- A legislative mandatory limit of iTFA in Australian foods was estimated to be a cost-effective strategy to reduce IHD and related deaths and healthcare costs.

- Socioeconomically disadvantaged groups and Australians outside major cities could potentially have the greatest health gains from such legislation.

- Our model estimates suggest that even in countries like Australia where intake is low, elimination of iTFAs can improve public health and health equality.

## Introduction

Ischemic heart disease (IHD) is the single most common cause of death in Australia, contributing to 12% of all deaths in 2016 [1]. A well-known dietary risk factor of IHD is intake of *trans*-fatty acids (TFAs), a group of unsaturated fatty acids with 1 or more double bonds in the *trans* configuration. TFA causes cardiometabolic dysfunction [2], and a meta-analysis of prospective studies reported that for every 2% of total energy from TFAs, IHD risk increased by 23% [3]. TFAs occur naturally at low levels in meat and milk from ruminants, but in most countries, intakes are predominantly driven by the use of partially hydrogenated vegetable oils in processed foods such as pastries [4].

Given the adverse effects of TFAs, the World Health Organization (WHO) recommends limiting TFA intake to 1 energy percent (%E), and countries have implemented strategies to reduce industrial-derived TFA (iTFA) in the food supply such as partially hydrogenated vegetable oils [5,6]. These policies range from voluntary reformulation, mandatory labelling, through to banning iTFA entirely. For example, iTFA content in foods must be ≤2% of total fat in Denmark [7], and in 2015, the United States Food and Drug Administration determined that partially hydrogenated oils are no longer considered generally recognized as safe for use in human food [8]. In Australia, voluntary reformulation, encouraged by the government and other health bodies, led to a reduced intake of iTFA mainly due to the reformulations of edible oil spreads by leading manufactures prior to 2000 [9,10]. However, between 2005 and 2013, there was no appreciable further reduction of iTFA in the Australian food supply [10]. According to the latest nationally representative dietary survey conducted in 2011 to 2012, Australians on average have intakes of TFA at 0.6% of daily energy, with an estimated 60% to 75% coming from ruminant (natural) sources and the rest being iTFA [10]. However, there is substantial variability around the average intake, with about 1 in 10 Australians estimated to exceed the WHO-recommended level of 1%E, especially those in the most socioeconomically disadvantaged groups [11]. Our prior modelling suggests that at a current level of intake, TFA continues to contribute to around 500 deaths per year in Australia, with the majority of these expected to accrue to those with less education and income [12].

A recent systematic review suggests that, not surprisingly, of all policy approaches currently implemented around the world to reduce iTFA, bans are the most effective [13]. Despite being comparatively low, the uneven distribution of consumption means that a part of the population consumes TFA in harmful quantities, and TFA intake levels in Australia continue to contribute to the IHD burden [12]. Further removal of iTFA from the food supply through mandatory limits (iTFA ban) could have additional benefits in Australia and other countries with similarly low TFA intake but no mandatory limits of iTFA content in foods. To our

knowledge, Australia has so far not considered a ban of iTFA. In order to further inform policy regulations related to iTFA in Australia, we conducted a modelling study that estimated the public health impact and cost-effectiveness of a national ban of iTFAs in Australia, considering both policy costs and reduced IHD burden and healthcare expenditures. Furthermore, we investigated whether the greatest benefits would be among the most disadvantaged groups of the Australian population. Apart from differences across socioeconomic groups, we estimated benefits separately in groups based on remoteness, given that Australians outside of metropolitan areas often have higher disease burden and poorer access to healthcare and diet compared to those living in major cities [14,15]. We hypothesized that a ban of iTFA in the food supply would be a cost-effective measure to reduce IHD burden and health disparities in Australia.

## Methods

This study is reported as per the Consolidated Health Economic Evaluation Reporting Standards (CHEERS) guideline (S1 Checklist). The initial study proposal is presented in S1 Text.

### Study design

We used a multiple cohort proportional multistate life table (Markov) model to estimate the impact on health outcomes and related costs of a complete ban of iTFAs from the Australian food supply. The ban would cover all foods and ingredients containing iTFA and therefore target both packaged and restaurant foods. Our model was constructed to calculate IHD-related outcomes and total healthcare costs resulting from the intervention [12,16]. The life table method transmits changes in iTFA intake to IHD-related morbidity and mortality in the modelled population. The analysis was conducted for the total Australian population and for pre-specified subpopulations: socioeconomic quintiles defined according to the Index of Relative Socio-Economic Disadvantage of the Socio-Economic Indexes for Areas (SEIFA) [17]; and inhabitants of major cities, inner regional Australia, and other regional areas (including outer regional, remote, and very remote Australia). We initially planned to evaluate the health impact of the ban stratified by Aboriginal status (S1 Text) but could not reliably estimate age- and sex-specific distributions due to small sample sizes. In all analyses, adults (age ≥20 y) were modelled in 5-year male and female cohorts, simulating each cohort until all individuals died or reached 100 years of age. Outcomes were compared between a reference population with TFA intake of the Australian population before the intervention and an intervention population with identical characteristics but lower TFA intake that is expected after the elimination of iTFAs from the food supply. The difference in health outcomes between reference and intervention populations were expressed in IHD incidence and deaths, life years, and health-adjusted life years (HALYs). Results were reported for 10-year and life span (i.e., the time from policy implementation until all individuals died or reached 100 years of age) time horizons. We used an 'extended' health sector perspective that included costs of legislation and industry, as these are directly related to the intervention. Costs were adjusted to 2010 values, and in line with the recommendations of the first and second panels on cost-effectiveness in health and medicine, we used a 3% discount rate in the main analysis [18,19].

### Data sources

**Intake of TFA in Australia.**   As previously described [12], nationally representative intakes of TFA were assessed using the 2011–2012 Australian National Nutrition and Physical Activity Survey (NNPAS) [20]. Baseline intake of TFA as %E was calculated per age and sex group in the total population; by remoteness of residence (major cities, inner regional areas, and outer regional areas); and by socioeconomic status in individuals belonging to each

socioeconomic quintile defined according to SEIFA (S1 Table). A mandatory ban in Denmark virtually eliminated iTFA from the food source [21]. Thus, we assumed the post-intervention TFA intake of all sex-age groups to be equal to the theoretical minimum risk distribution, i.e., 0.50 ± 0.05%E in our primary model (**Table 1**), based on the likely average intake of nonindustrial-derived TFAs (i.e., from meat and dairy from ruminant animals) [12,22].

**Health outcomes.** Estimates of total and subgroup-specific population demographics, mortality rates, and IHD burden (prevalence, incidence, and mortality) were based on data from the Australian Bureau of Statistics, the Australian Institute of Health and Welfare (AIHW), and the Global Burden of Disease project (**Table 1**). Our model included adults aged 20 to 100 y; 51% of the total model population were women, 68% lived in major cities, 18% in inner regional Australia, and 11% in outer regional, remote, or very remote areas. Age-specific relative risks of TFA intake and IHD were based on meta-analyses of findings from prospective cohort studies [23]. Children and adolescents (age <20 y) were not included in the model, given the low IHD burden and the lack of well-established relative risks of TFA intake and IHD in that age group.

**Healthcare and policy costs.** Healthcare costs related to IHD treatment including hospital services, out-of-hospital medical services, pharmaceuticals, and health professionals were based on Disease Costs and Impact Study (DCIS) 2001 data from the AIHW, inflated to 2010 prices using AIHW health price inflation values [24]. Healthcare costs for diseases and injuries unrelated to IHD due to additional years of life gained were also taken from the DCIS.

The policy costs estimated for an elimination of iTFA in Australia included costs for government (i.e., legislation and monitoring costs) and industry (i.e., initial and ongoing reformulation as well as product labelling costs). Legislation costs were estimated as described in detail by Lal and colleagues, using a costing framework for public health legislation including parliamentarians' time, annual expenses for the House of Representatives and the Senate, legislation drafting and publication, and policy advice [25]. In the absence of Australian estimates for TFA monitoring and reformulation costs, we utilized United Kingdom estimates (annual cost of £2.4 million [26]) to calculate equivalent Australian dollar costs of TFA monitoring, while adjusting for population size differences between the countries (i.e., multiplying with the ratio of 0.35, derived from dividing the Australian (22.3 million) by the UK population (63.3 million)). Reformulation costs were calculated using equivalent Australian dollar costs from UK estimates (£25,000 per product [26]) multiplied by the number of products in the Australian food supply potentially containing iTFA. We estimated this number by identifying products in the 2018 Australian FoodSwitch database [27] that contained any terms indicative of iTFA in the ingredient list (i.e., 'partially hydrogenated fat', 'hydrogenated vegetable oil', or 'hydrogenated') [28]. Of 28,349 foods included in the analysis, 131 products (0.5%) contained specific ingredients indicative of iTFA [28]. As the intervention could lead to increased costs to industry for changes in packaging and loss through disuse of existing packaging, we estimated a 1-time repackaging cost as 10% of reformulation costs [26]. In line with previous modelling studies [26,29], we assumed an ongoing annual industry cost of equalling 1% of the initial reformulation cost to conservatively account for reduced industry profits.

## Statistical analysis

**Estimation of health benefits and cost-effectiveness.** The reference and intervention TFA intakes and the relative risk of IHD per %E of TFA intake were used to calculate the potential impact fraction (PIF) for estimation of the proportional change in IHD incidence due to the elimination of iTFAs (Eq (1)). Barendregt's continuous 'distribution shift' PIF

**Table 1. Data sources for modelling.**

| Input | Stratification | Values | Source | Note |
|---|---|---|---|---|
| Pre-intervention TFA intake, %E | Age, sex | See S1 Table | 2011–2012 NNPAS | For subgroups[1], weighted mean and standard deviations from 2011–2012 NNPAS were assumed to approximate nationally representative subgroup-specific means and standard deviations intake. |
| Post-intervention TFA intake, %E | n/a | Mean ± SD: 0.50 ± 0.05 (primary analysis) Mean ± SD: 0.4 ± 0.04 (sensitivity analysis) | Wu et al., Nutrients, 2017 Allen et al., BMJ, 2015 | |
| Theoretical minimum risk distribution of TFA intake, %E | n/a | Mean ± SD: 0.50 ± 0.05 (primary analysis) Mean ± SD: 0.4 ± 0.04 (sensitivity analysis) | Wu et al., Nutrients, 2017 | |
| RR for CHD per 2%E from TFA | Age | 25–34 y: 1.42 (1.28–1.57) 35–44 y: 1.40 (1.27–1.54) 45–54 y: 1.33 (1.22–1.45) 55–64 y: 1.27 (1.18–1.36) 65–74 y: 1.22 (1.15–1.29) ≥75 y: 1.16 (1.11–1.21) | Wang et al., JAHA, 2017 | For each model iteration, random draws from age-specific lognormal RR distributions were made. |
| Population size | Age, sex | See S2 Table | ABS report 3101.0 –Australian Demographic Statistics, December 2010 –TABLE 59. Estimated Resident Population By Single Year Of Age, Australia. | |
| Mortality rate | Age, sex | See S3 Table | ABS report 3302.0 –Deaths, Australia, 2010 – TABLES 4.1–4.2[1]. | |
| IHD incidence, prevalence, and case fatality | Age, sex | See S4–S6 Tables | Global burden of disease project 2010 | |
| Subgroup population sizes* | Subgroup, age, sex | See S2 Table | ABS *2011 Census of Population and Housing*, TableBuilder. Findings based on the use of ABS TableBuilder data. | |
| Subgroup[2] mortality rates* | Subgroup[2], age, sex | See S3 Table | Mortality inequalities in Australia 2009–2011 Table S3.1: Deaths by socioeconomic group, by sex, and by age group, 2009–2011 | Mortality rates in each remoteness or socioeconomic subgroup were available per sex in 5 specific age groups (15–24 y, 35–44 y, 45–64 y, 65–84 y, and 85+). Mortality rates for each subgroup-, sex-, and year of age-stratum was calculated by multiplying sex- and year of age-specific mortality rate in the total population with the ratio of the subgroup-specific mortality rate and the age-adjusted mortality rate of the total population. |
| Subgroup[2] IHD incidence and prevalence* | Subgroup[2], age, sex | See S4 and S5 Tables | AIHW Cardiovascular disease web pages data tables[3] | IHD incidence for each subgroup-, sex-, and age-stratum was calculated by multiplying sex-age-specific IHD incidence in the total population with the ratio of the subgroup-specific CVD hospitalisation rate and the age-adjusted CVD hospitalisation rate of the total population. Subgroup-, sex- and age-specific IHD prevalence was calculated similarly. |
| Disability weights | Sex | Male: 7.67% Female: 7.63% | Global burden of disease project 2010[4] | Weighted average of MI, angina, and heart failure |
| Healthcare costs | Sex, age | See S5 Table | AIHW 2001 inflated to 2010 | Total and IHD-related healthcare costs were estimated separately. |

*(Continued)*

**Table 1.** (Continued)

| Input | Stratification | Values | Source | Note |
|---|---|---|---|---|
| Legislation costs | n/a | Mean ± SD: 1,090,000 ± 77,497 AUD | Lal et al., PLoS Med., 2017 | For each model iteration, a random draw from a gamma distribution of legislation costs was made. |
| Monitoring cost | n/a | 1,557,422 AUD/year | Allen et al., BMJ, 2015 | Monitoring costs were estimated using equivalent Australian dollar costs (2011 exchange rate: 1.84 AUD/GBP) of UK estimates (annual cost of 2.4 million GBP), while adjusting for 2011 population size differences between the UK (63.3 million) and Australia (22.3 million). |
| Initial industry reformulation | n/a | 6,010,391 AUD (primary model) 12,020,783 AUD (sensitivity model) | Allen et al., BMJ, 2015 and Huang et al., ANZJPH, 2020 | Reformulation costs were calculated using equivalent Australian dollar costs (2011 exchange rate: 1.84 AUD/GBP) from UK estimates (25,000 GBP per product) multiplied by the number of products in the Australian food supply potentially containing iTFA (primary model: $n = 131$; sensitivity analysis: $n = 262$). |
| Annual industry reformulation | n/a | 60,104 AUD/year (primary model) 120,208 AUD/year (sensitivity model) | Allen et al., BMJ, 2015 | Annual cost to industry equalling 1% of the initial reformulation cost was assumed. |
| Initial industry repackaging | n/a | 601,039 AUD (primary model) 1,202,078 AUD (sensitivity model) | Allen et al., BMJ, 2015 | Increased costs to industry for changes in packaging and loss through disuse of existing packaging were estimated as a 1-time cost to 10% of reformulation costs. |

ABS, Australian Bureau of Statistics; AIHW, The Australian Institute of Health and Welfare; AUD, Australian dollar; CHD, coronary heart disease; CVD, cardiovascular disease; %E, energy percentage; GBP, British pound; IHD, ischemic heart disease; iTFA, industrial TFA; NNPAS, National Nutrition and Physical Activity Survey; RR, relative risk; SD, standard deviation; SEIFA-IRSD, Socio-Economic Indexes for Areas–Index of Relative Socio-Economic Disadvantage; TFA, *trans*-fatty acid; UK, United Kingdom.

[1]URL: https://www.ausstats.abs.gov.au/ausstats/subscriber.nsf/0/F773BDC941982EF5CA257943000CFD5D/$File/33020_2010.pdf (Accessed September 10, 2019).

[2]Subgroups includes: socioeconomic status (defined by SEIFA-IRSD quintiles), and remoteness (major city; inner regional, and outer regional/remote/very remote).

[3]URL: https://www.aihw.gov.au/getmedia/0c4fd299-edc6-40f5-b5e1-69d48c9c9648/cvd-ccc2016-20409.xls.aspx (Accessed July 25, 2019).

[4]Salomon *et al.*, Lancet., 380(9859):2129–2143.

method was used [30].

$$\text{PIF}_{as} = \frac{\int_{x=0}^{m} RR_a(x)P_{as}(x)dx - \int_{x=0}^{m} RR_a(x)P'_{as}(x)dx}{\int_{x=0}^{m} RR_a(x)P_{as}(x)dx} \tag{1}$$

The $\text{PIF}_{as}$ is the potential impact fraction for age group a and sex s, $RR_a(x)$ is the relative risk as a function of the exposure x (i.e., TFA intake), $P_{as}(x)$ is the reference TFA intake distribution, and $P'_{as}(x)$ is the intervention TFA intake distribution. The PIF was used to calculate the effect on IHD incidence due to the reduction in TFA intake (Eq (2)).

$$I' = I(1 - PIF) \tag{2}$$

*I* is the IHD incidence in the reference population, *I'* is the IHD incidence in the intervention population, and PIF is the potential impact fraction. The estimated incidence rates were used in life tables to calculate reference and intervention IHD prevalence and mortality. To account for time spent in suboptimal health due to IHD and any other conditions present, we calculated HALYs. Each year of life lived was adjusted using all-cause 'prevalent years lived with disability' values from the Global Burden of Disease 2010 study. These sum the loss of disease-related quality of life by age based on prevalence and disability weight for all conditions

(Table 1). One HALY thus represents the equivalent of a year in perfect health. HALYs gained were calculated as the difference in HALYs between the reference and intervention populations. Changes in healthcare expenditures were estimated both for IHD-related healthcare and total healthcare. The change in IHD-related healthcare expenditure was based on the predicted reduction in IHD mortality and morbidity. Overall healthcare costs in added years of life were also included [31].

Net costs included policy costs and healthcare costs (including costs unrelated to IHD) and were utilized to calculate the incremental cost-effectiveness ratios (ICERs), defined as the difference in net costs of the intervention compared to current practice, divided by the difference in HALYs. We initially planned to use WHO benchmarks for definition of cost-effectiveness (S1 Text) but decided to use an Australia-specific definition instead where cost-effectiveness was defined as ICER < Australian dollar (AUD) 169,361 per HALY gained (i.e., the value of a statistical life year in 2007 [32], inflated to 2010 using consumer price index [33]). Cost saving was defined as a negative net cost.

## Subgroup analyses

For each prespecified subgroup (i.e., SEIFA quintiles and areas of residence), subgroup-specific models were created by substituting key input parameters (i.e., IHD incidence, mortality rate, TFA intake, and population number) with subgroup-specific data (Table 1).

Health inequalities were estimated by comparing HALYs gained in extreme SEIFA quintiles and by using SEIFA quintiles to calculate concentration indices [34]. The concentration index quantifies the distribution of HALY over socioeconomic quantiles and takes a negative value when HALYs gained are disproportionally concentrated in the most disadvantaged groups [34].

## Uncertainty and sensitivity analysis

The parameter uncertainty around the modelled estimates was quantified using Monte Carlo simulations ($n$ = 2,000). For each iteration, a draw was made from the distributions of TFA intake, relative risks, and legislation costs. The point estimate and 95% uncertainty intervals (UI) were defined as the 50th and 2.5th to 97.5th percentiles, respectively, of the distribution of the intervention effects (e.g., HALYs gained) estimated across all 2,000 iterations using the Ersatz version 1.35 software (Epigear International, Sunrise Beach, Australia). Similarly, Monte Carlo simulations ($n$ = 2,000) of policy costs were conducted in RStudio version 1.1.423 (Boston, Massachusetts, USA).

Univariate sensitivity analysis was used to explore the impact of variation in discount rates (0% and 6%), theoretical minimum risk distribution, and TFA exposure. We assumed a lower post-intervention intake and theoretical minimum risk distribution, both with mean of 0.4%E [26], i.e., the estimated maximum level of TFA from ruminant sources in the UK, and 0.04%E as standard deviation (i.e., 10% of the mean). We also evaluated the impact of higher post-intervention intakes (e.g., due to suboptimal compliance to the ban or lower contribution of iTFA to total TFA) with greater mean (0.52%E to 0.60%E) and with standard deviations equal to 10% or 50% of the mean. We also evaluated the impact of lower pre-intervention intakes, with the post-intervention mean intake of the primary model (i.e., 0.50%E) but only half the difference between the pre- and post-intervention mean intakes. In order to test the effect of a potentially greater prevalence of iTFA in Australian foods (e.g., due to TFA-containing restaurant foods not included in the FoodSwitch database), the number of products potentially containing iTFA was assumed to be twice as many as identified in the FoodSwitch database [28]. Experience of TFA regulations in Denmark has suggested negligible reformulation costs [29],

and thus we conducted a sensitivity analysis assuming no industry costs. Given the differences between the UK (from where our assumption for the primary model was based) and Australia (e.g., geographic dispersion), it is possible that monitoring costs maybe greater per capita in Australia compared to the UK. Thus, we also evaluated the impact of 25% greater monitoring costs compared to our primary model.

## Results

### Overall health gains, costs, and cost-effectiveness

Compared to current levels of TFA intake, a ban of iTFAs was estimated to avert 2,294 IHD deaths and 9,931 IHD events during the first 10 years, which over the population lifetime (i.e., the time from policy implementation until all individuals died or reached 100 years of age) would amount to 41,877 averted IHD deaths and 56,759 averted IHD events (**Fig 1** and **Table 2**). The ban was estimated to result in 6,377 HALYs and 5,532 total life years gained over 10 years; over the population lifetime, 108,321 HALYs and 127,268 total life years would be gained (**Fig 2** and Table 2). During the first 10 years, the per capita total and IHD-related healthcare costs savings were estimated as AUD 20 and AUD 15 per person, respectively (**Fig 3A** and Table 2). Over the population lifetime, AUD 22 IHD-related healthcare cost and AUD 23 total healthcare cost would be saved per person (**Fig 3B** and Table 2).

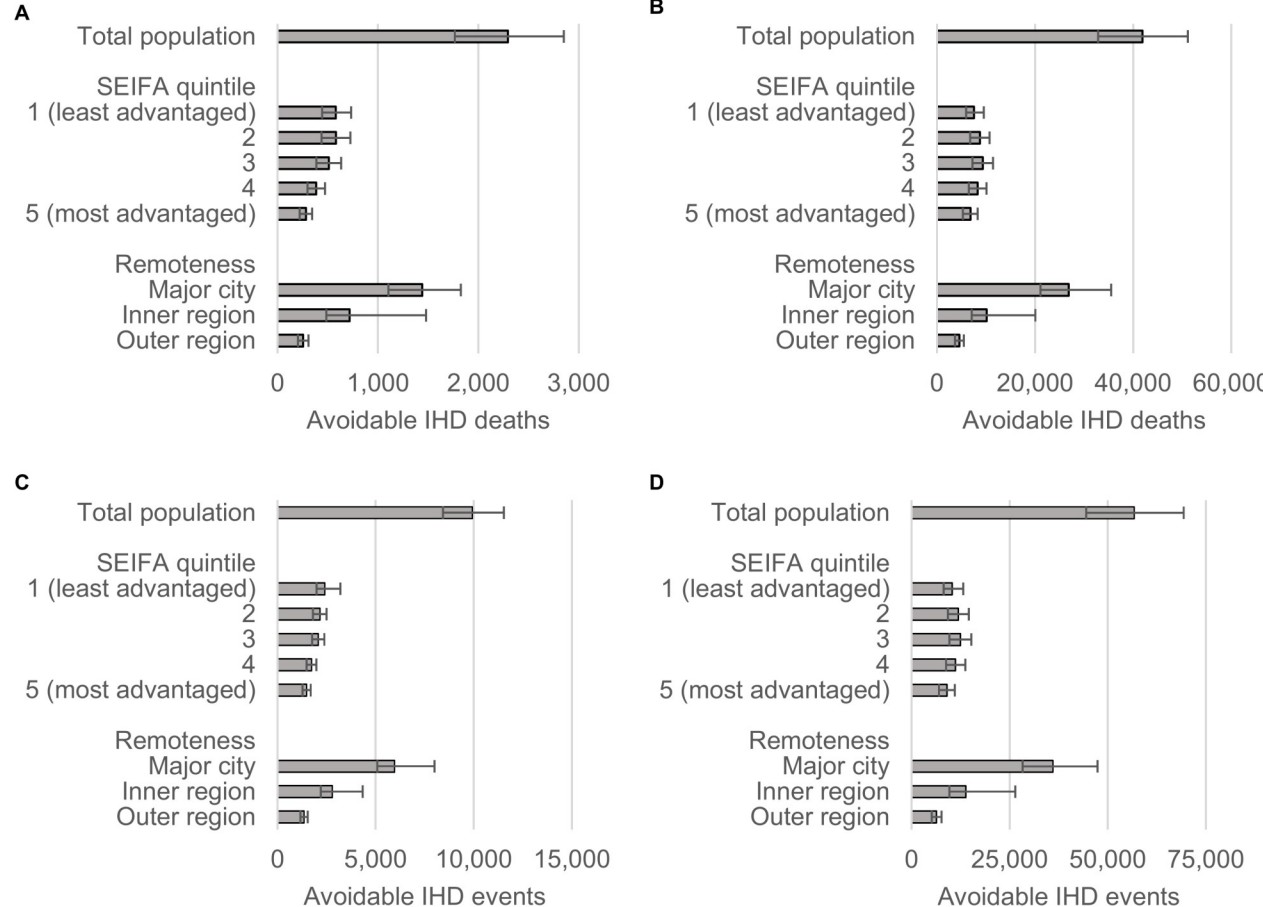

**Fig 1. Ischemic heart deaths (A–B) and incidences (C–D) avoidable during the first 10 years (A, C) and over the population lifetime (B, D).** Error bars represent 95% uncertainty intervals. IHD, ischemic heart disease; SEIFA, Socio-Economic Indexes for Areas.

**Table 2. Estimated health gains, costs, and cost-effectiveness of eliminating industrial TFA from the Australian food supply over 10 years and over the population lifetime.**

| | 10 years Estimate (95% UI) | Lifetime Estimate (95% UI) |
|---|---|---|
| IHD events avoided | 9,931 (8,429; 11,532) | 56,759 (44,493; 69,358) |
| IHD deaths avoided | 2,294 (1,765; 2,851) | 41,877 (32,855; 51,138) |
| Life years gained | 5,532 (4,285; 6,845) | 127,268 (106,176; 148,917) |
| HALYs gained | 6,377 (5,151; 7,685) | 108,321 (91,187; 125,986) |
| Change in IHD-related healthcare costs, million AUD | 80 (-98; -69) | -538 (-628; -463) |
| Per capita, AUD | -15.0 (-18.3; -11.9) | -21.7 (-26.0; -17.3) |
| Change in total healthcare costs, million AUD | -14 (-34; 3) | 157 (48; 260) |
| Per capita, AUD | -19.7 (-24.5; -15.2) | -23.0 (-28.8; -17.5) |
| Policy costs, million AUD | 21.5 (21.3; 21.6) | 56.0 (55.9; 56.2) |
| Governmental costs | 14.3 (14.2; 14.5) | 47.6 (47.4; 47.7) |
| Legislation | 1.1 (0.9; 1.2) | 1.1 (0.9; 1.2) |
| Monitoring | 13.3 | 46.5 |
| Industry costs | 6.6 | 7.8 |
| Initial reformulation | 6.0 | 6.0 |
| Annual industry costs | 0.5 | 1.8 |
| Repackaging | 0.6 | 0.6 |
| Net costs[1], million AUD | 7.2 (-12.9; 24.4) | 212.9 (104.1; 315.8) |
| ICER[2], AUD/HALY | 1,153 (dominant; 3,578) | 1,961 (1,015; 2,756) |

AUD, Australian dollar; HALY, health-adjusted life year; ICER, incremental cost-effectiveness ratio; IHD, ischemic heart disease; TFA, *trans*-fatty acid; UI, uncertainty interval.

[1]Calculated as the sum of policy costs and change in total healthcare costs.

[2]Calculated as net costs divided by HALYs gained.

Banning and elimination of iTFA from the Australia food supply was estimated to cost nearly AUD 22 million during the first 10 years and AUD 56 million over the population lifetime (Table 2), with a majority of estimated costs attributed to government costs for monitoring. During the same time periods, the IHD-related healthcare cost savings, compared to no ban, were estimated to AUD 80 million and AUD538 million, respectively (Table 2). When other healthcare costs (i.e., including those related to a greater and older population resulting from the estimated reduced IHD mortality) were considered, the savings were estimated to AUD 14 million over 10 years. However, due to reduced IHD mortality resulting in a larger and older population, it was estimated that the ban would result in AUD157 million additional healthcare costs over the population lifetime (Table 2). Thus, the net policy costs over 10 years and the population lifetime were estimated as AUD 7.2 million and 212.9 million, respectively (Table 2). The elimination of iTFA in Australia was estimated to be cost saving (i.e., net cost <0 AUD) to highly cost-effective during the first 10 years (1,153 AUD/HALY [95% UI: dominant; 3,578], i.e., ≤2% of the value of a statistical life year) and highly cost-effective over the population lifetime (1,961 AUD/HALY [95% UI: 1,015; 2,756], Table 2).

## Effect on health disparities

Over the first 10 years, the iTFA ban, compared to current TFA intake, was estimated to reduce health inequalities, and the most socioeconomically advantaged quintile had the lowest proportions of averted IHD deaths (12%) and events (15%); and HALY (14%), as well as life years gains (12%) compared to other quintiles (Figs 1 and 2). The greatest proportion of HALYs would accrue to the least advantaged quintiles (Fig 4 and Table 3). For example, the

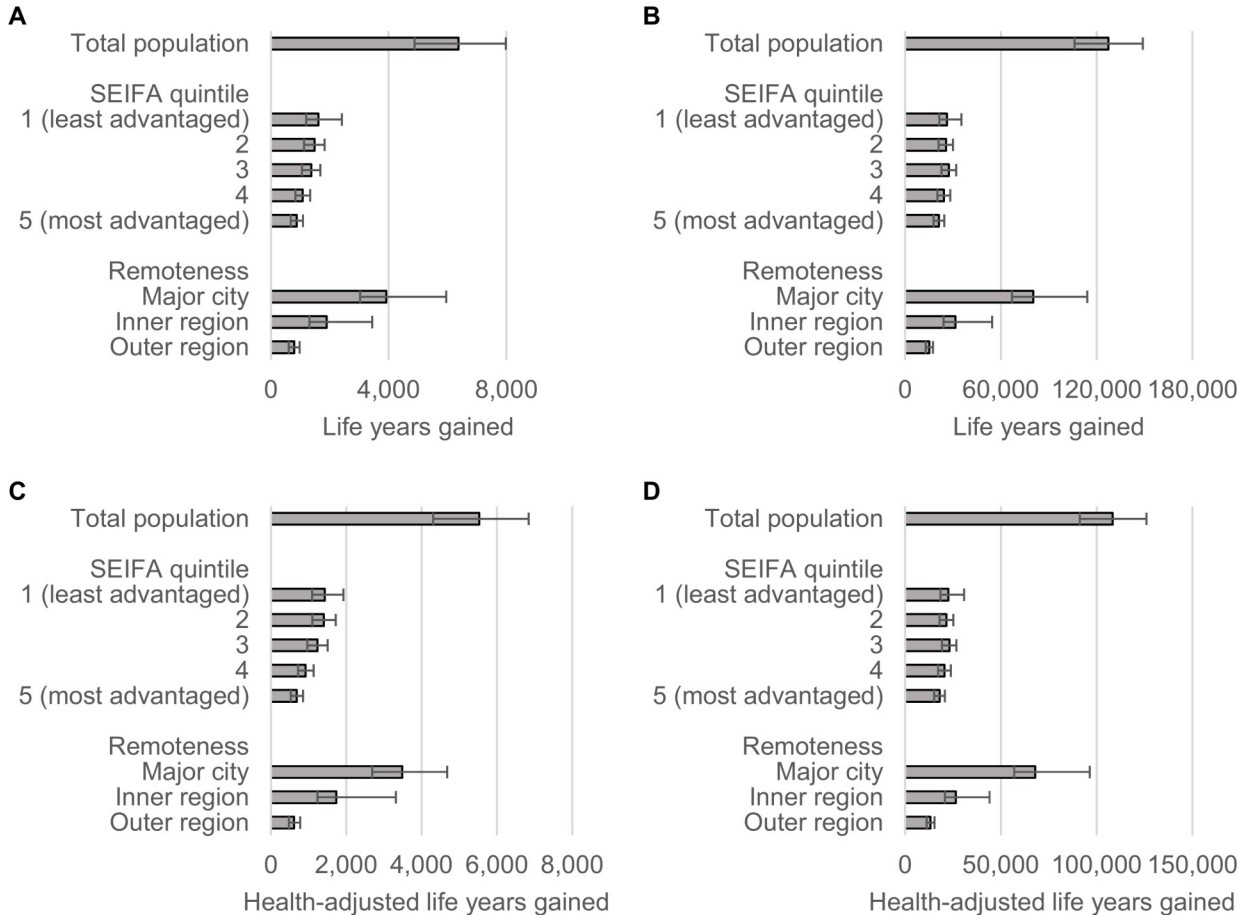

**Fig 2. Total (A–B) and HALYs (C–D) gained during the first 10 years (A, C) and over the population lifetime (B, D).** Estimates are presented separately for the total population and selected subgroups. Error bars represent 95% uncertainty intervals. HALYs, health-adjusted life years; SEIFA, Socio-Economic Indexes for Areas.

most disadvantaged quintile of the population would gain 742 more HALYs over 10 years, compared to the most advantaged quintile (Table 3). Similar reductions in health disparity were observed for healthcare cost savings (Fig 3). Over the population lifetime, the ban was not estimated to significantly reduce health disparities (Fig 4 and Table 3). Although only 30% of Australians live outside major cities, around 4 in 10 averted IHD events and deaths, as well as, HALYs and life years gained were estimated to accrue to this group (Figs 1 and 2). In addition, they would experience greater healthcare cost saving per capita compared to those living in major cities (Fig 3).

## Sensitivity analyses

For each of the deterministic sensitivity analyses, the iTFA ban was estimated to be cost saving or highly cost-effective (**Fig 5** and S7 Table), with no 95% UI exceeding 20% of the value of a statistical life year. The greatest ICER during the first 10 years was estimated when assuming 0% discount rate: 3,308 AUD/HALY (95% UI: dominant; 5,970). Assuming the same discount rate over the population lifetime also suggested that the ban would be highly cost-effective, with estimated ICER 15,423 (95% UI: 14,176; 16,540). On the contrary, when assuming a 6% discount rate, the ban was estimated to be cost saving (Fig 5). Assumptions regarding lower

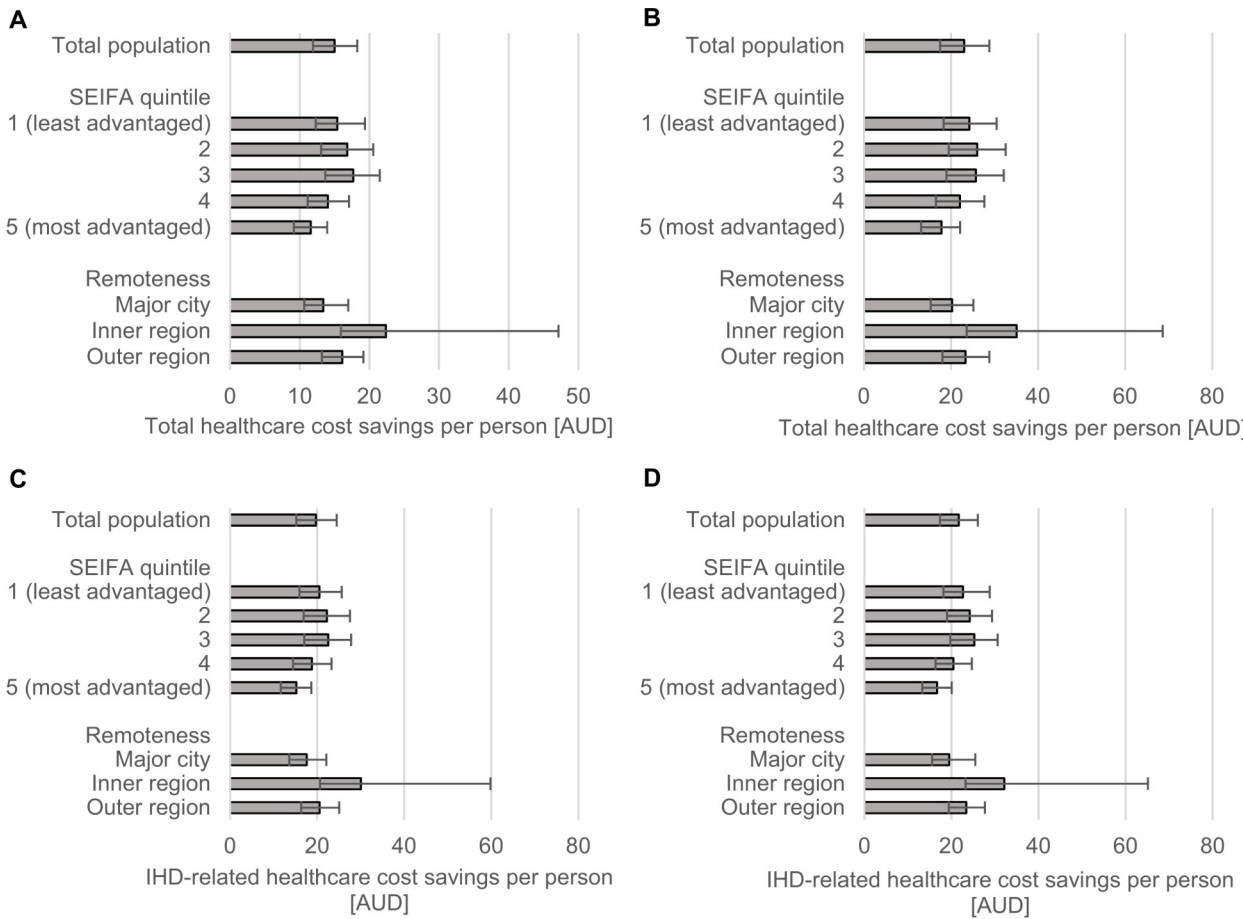

**Fig 3. Total (A–B) and IHD-related (C–D) healthcare cost savings per capita estimated over 10 years (A, C) and population lifetime (B, D).** Estimates are presented separately for the total population and selected subgroups. Error bars represent 95% uncertainty intervals. AUD, Australian dollar; IHD, ischemic heart disease; SEIFA, Socio-Economic Indexes for Areas.

pre- or post-intervention TFA intakes, abundance of iTFA-containing products in Australia, and monitoring or industry costs had minor impact on model estimates (Fig 5). Assumptions of a less effective elimination of iTFA (i.e., greater post-intervention TFA intake with wider distribution) resulted in increased ICER, especially over the 10-year horizon (S7 Table). However, the estimated ICER across all sensitivity analyses were estimated to be <20% of the cost-effectiveness threshold, i.e., the intervention remained highly cost-effective.

## Discussion

We used national representative data in Markov cohort models to estimate the impact on IHD burden and health equity as well as the cost-effectiveness of a national ban of iTFA in Australia compared to no ban and current levels of TFA intake. Our model estimated that elimination of iTFA from the Australian food supply could prevent around 2,000 deaths and 10,000 incident IHD-events over the first 10 years, with greater benefits among socioeconomically disadvantaged groups and Australians outside major cities. Across the population's lifetime (i.e., the time from policy implementation until all individuals died or reached 100 years of age), an iTFA ban was estimated to avert nearly 42,000 IHD deaths. The intervention was estimated to be cost saving or cost-effective over the initial 10 years as well as over the population lifetime.

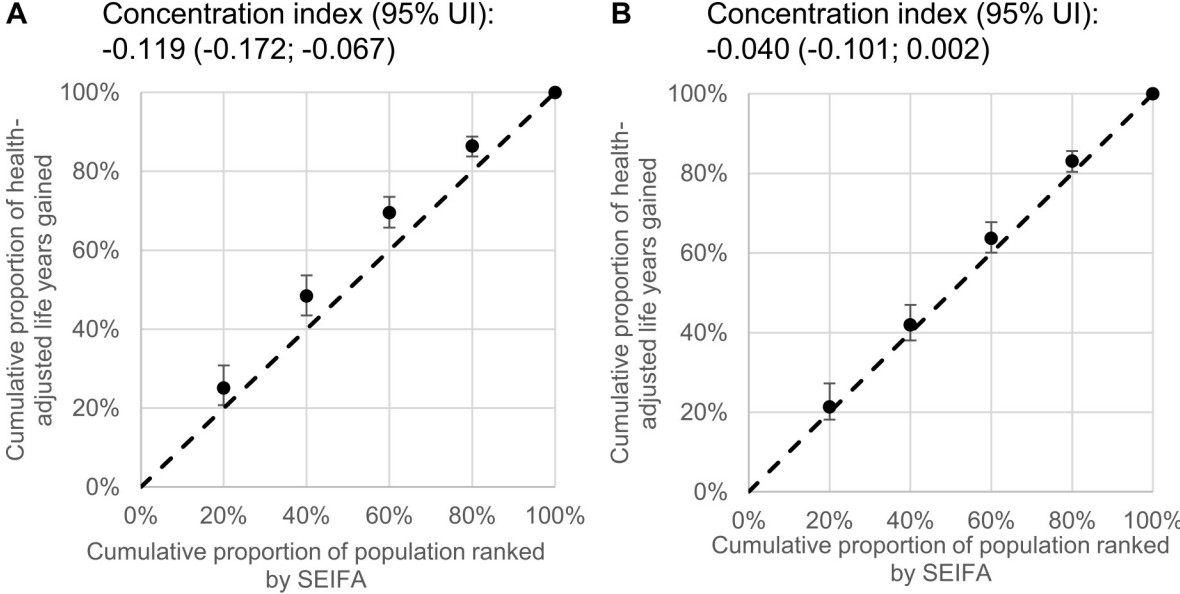

**Fig 4. Relative inequality based on socioeconomic status in HALYs gained estimated over 10 years (A) and population lifetime (B).** Values above the dotted unity lines represent greater benefits among disadvantaged groups (i.e., lowest SEIFA ranking). Dots and error bars indicate median and 2.5–97.5 percentiles of $n$ = 2,000 estimates. The concentration index quantifies the distribution of HALY over socioeconomic quantiles and takes a negative value when HALYs gained are disproportionally concentrated in the most disadvantaged groups. HALY, health-adjusted life years; SEIFA, Socio-Economic Indexes for Areas; UI, uncertainty interval.

## Interpretation and policy implications

Previous modelling studies have estimated the impact on cardiovascular disease burden of plausible legal limits or bans of iTFA in the UK [26,29,35,36] and the EU [37]. In all of these studies, such legislative policies were estimated to reduce the number of cardiovascular events or deaths and/or increase HALYs or life years [38]. Three of the studies evaluated the cost-effectiveness of bans or legal limits, taking into consideration health gains, healthcare cost

**Table 3. HALYs gained by socioeconomic quintiles, absolute difference between extreme quintiles, and concentration index.**

| | 10 years Estimate (95% UI) | Lifetime Estimate (95% UI) |
|---|---|---|
| SEIFA-IRSD quintiles, HALYs gained | | |
| 1 (most disadvantaged) | 1,614 (1,276; 2,107) | 22,512 (18,458; 30,830) |
| 2 | 1,485 (1,181; 1,801) | 21,528 (17,914; 25,147) |
| 3 | 1,369 (1,099; 1,639) | 23,105 (19,252; 26,771) |
| 4 | 1,081 (881; 1,294) | 20,486 (17,202; 23,833) |
| 5 (most advantaged) | 876 (714; 1,041) | 18,014 (15,178; 20,776) |
| Difference between first and fifth quintiles, HALYs gained | 742 (357; 1,241) | 4,648 (-491; 12,910) |
| Concentration index[1] | -0.119 (-0.172; -0.067) | -0.040 (-0.101; 0.002) |

HALY, health-adjusted life year; SEIFA-IRSD, Socio-Economic Indexes for Areas–Index of Relative Socio-Economic Disadvantage; UI, uncertainty interval.

[1]The concentration index quantifies the distribution of HALY over socioeconomic quantiles and takes a negative value when HALYs gained are disproportionally concentrated in the most disadvantaged groups.

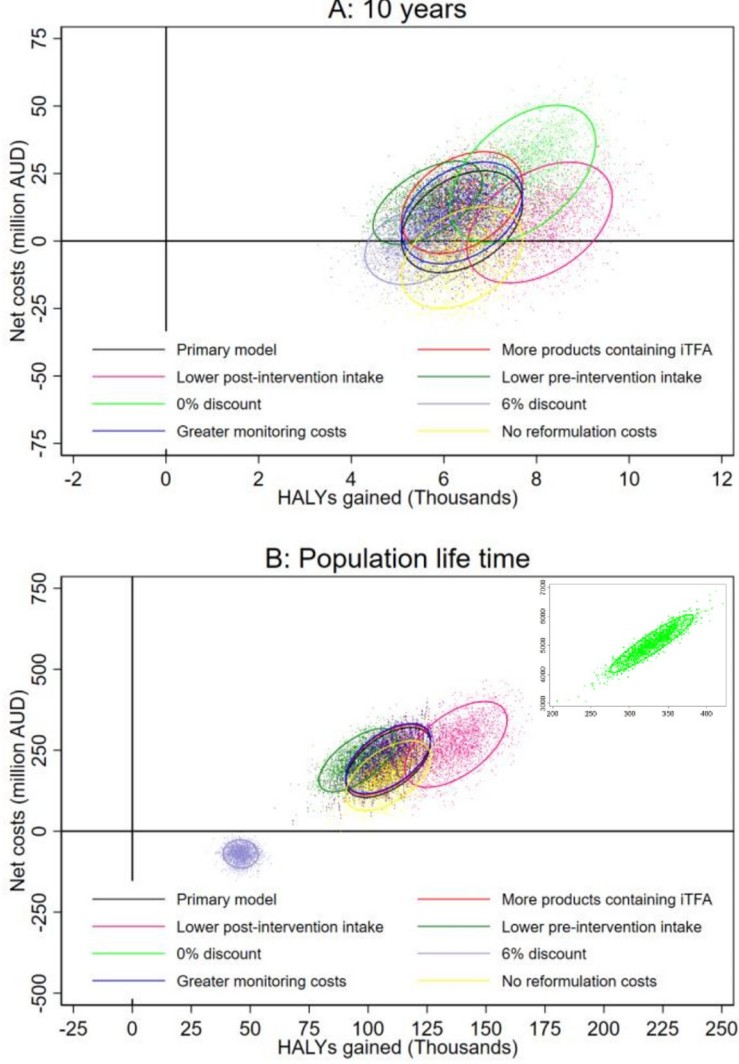

**Fig 5. Net costs and HALYs gained during the first 10 years (A) and over the population lifetime (B) estimated in the primary model and in deterministic sensitivity analyses.** See Methods for descriptions of sensitivity analyses. Dots represent estimates of *n* = 2,000 simulations, and ellipses indicate 95% confidence intervals centred around their means. AUD, Australian dollar; HALY, health-adjusted life years; iTFA, industrial TFA.

savings, and policy costs [26,29,37]. A legal limit of iTFA in the EU (assumed to remove all iTFA) [37] and total bans in the UK [26,29] were estimated to be cost saving. Bans of iTFA in the UK were also estimated to reduce health inequalities [26,29].

An important and novel aspect of our study is that the modelled population already has a low average level of TFA intake (mean: 0.59%E). In previous modelling studies of different strategies to reduce TFA intake in the UK and the EU [26,37], where average TFA intake is higher (e.g., estimated TFA intake in the UK: 0.79%E [26]), it has been estimated that total bans of iTFA across the food supply would be superior to other evaluated strategies (e.g., voluntary limits and mandatory labelling or TFA ban targeting restaurant and takeaway food) with regard to health benefits and cost-effectiveness [26,37]. Our study builds on and significantly extend these prior studies and suggest legislating to remove iTFA could still lead to

significant health benefits and be done in a highly cost-effective way, even in a population with relative low average TFA intake like Australia. These findings support the WHO's call to eliminate TFA from the food supply around the world and strengthen the case that such a move likely represents a public health 'best-buy' [6]. A ban of iTFA may be particularly beneficial in certain Australian subpopulations where high TFA intakes (approximately 3%E) are possible [12], and our modelling results comparing estimated benefits accrued among groups of different social-economic circumstances and areas of residence support this assertion. These findings are consistent with prior estimations that an iTFA ban would improve health equity [26] due to the social economic gradient in iTFA consumption and burden of IHD. Experiences from other countries suggest that legislating or other policy mechanisms to eliminate iTFA is feasible. In Denmark, iTFA have been virtually eliminated after legislation introduced nearly 2 decades ago that limits the TFA content in food [7]. In the US, the Food and Drug Administration no longer considers partial hydrogenated oils to be generally recognized as safe (and thereby not allowed in foods) [8]. Major food companies have also recently pledged to remove iTFA from their products [39]. However, objective evaluations of similar voluntary actions by the food industry suggest such commitments often fail or only partially achieve their goals [40], and therefore, other mechanisms such as government legislation to ban iTFA may still be required. An independent review on food labelling, commissioned by the Australian Government in 2011, specifically recommended mandatory labelling of TFA if iTFA in foods were not phased out by 2013 [41]. However, no such regulations have been implemented.

## Strengths

Data used in our modelling were derived whenever possible from nationally representative sources, increasing validity, and generalizability of the results. This include the use of data from the most up-to-date, individual-level, nationally representative NNPAS. Data from a meta-analysis of prospective studies directly linking consumption of TFA to incidence of IHD were used instead of modelling a link mediated by blood lipids, and thus include potential non-lipid-mediated effects of TFA intake such as inflammation, which may also account for the harmful effect of TFA on IHD [42]. Healthcare costs estimated included both IHD-related and other healthcare costs and thus allowed estimation of changes in total healthcare expenditures. We estimated the health benefits and healthcare costs of eliminating iTFA among socioeconomic and urban–rural subgroups. We used SEIFA, which compared to income alone, is a more comprehensive measure of relative socioeconomic advantage and disadvantage in Australian population based on a wide range of socioeconomic indicators such as income, education, employment, occupation, and housing, and thus may better assess social economic status [43].

## Limitations

The most recent nationally representative assessment of TFA intake was conducted nearly a decade ago, and iTFA intake may have changed. However, there has been no government policy or mandate to reduce iTFA in the last 2 decades [10]; iTFA content in food products has been largely similar between 2005 and 2013 [10], and TFA intake remained stable from 1995 to 2012 [4]. This suggests that it is reasonable to assume fairly stable TFA intake levels since 2012. The number of products potentially containing iTFA were identified using the 2018 FoodSwitch database, and it is possible that the number of identified products would be greater if assessed concomitantly with the NNPAS in 2012. However, in the absence of policy and active government engagement with food industry in Australia, there is little reason to suspect that the number of products with iTFA has changed substantially in the food supply over

recent years, and in sensitivity analyses, doubling the number of products with iTFA did not materially impact our findings. The sample size of some age and sex groups was small (e.g., Australian outside major cities and inner regional areas), which may have resulted in imprecisely estimated means and standard deviations of TFA intake, especially in the subgroups analyses.

Consistent with prior modelling papers, our model uses risk estimates of change in TFA against the overall diet rather than specific substitution with other types of fats. If the iTFA would be systematically replaced by saturated fatty acids, the impact of the ban could be lower than estimated here. However, evidence suggest no overall increase in saturated fatty acid content in food products after previous reductions of iTFA [10]. Indirect costs, e.g., productivity loss due to absenteeism or disability, were not included in the estimation of healthcare cost savings, which means that the societal savings from the intervention are likely to be substantially underestimated. When Australia-specific cost data were not available, we used costing frameworks from the UK and New Zealand, which may under or overestimate such costs. Our modelling study does not prove that a ban of iTFA will prevent IHD; rather, it provides important quantitative estimates, corresponding uncertainty, and assessments of sensitivity of the findings to different inputs, resulting in a range of plausible effects on IHD burden and cost-effectiveness of legislating an iTFA ban in Australia to help inform policy makers.

## Conclusions

Our model estimates suggest that a ban of iTFAs, compared to no ban and current TFA intake, could be a highly cost-effective strategy to reduce the Australian IHD burden and could lead to tens of thousands of prevented premature deaths. Introducing such a policy was also estimated to reduce social–economic and urban–rural inequalities in IHD disease burden over the first 10 years.

## Supporting information

**S1 Checklist. CHEERS checklist.**
(DOCX)

**S1 Text. Study proposal.**
(DOCX)

**S1 Table. Total and subgroup-specific *trans*-fatty intakes (%E) per age group, estimated in the 2011–2012 National Nutrition and Physical Activity Survey (NNPAS).**
(DOCX)

**S2 Table. Total and subgroup-specific population size per year of age.**
(DOCX)

**S3 Table. Total and subgroup-specific mortality rate (deaths per 100,000) per year of age.**
(DOCX)

**S4 Table. Total and subgroup-specific IHD incidence (%) per year of age.**
(DOCX)

**S5 Table. Total and subgroup-specific IHD prevalence (%) per year of age.**
(DOCX)

**S6 Table. IHD case fatality (%) per sex and year of age.**
(DOCX)

**S7 Table. Cost-effectiveness of eliminating industrial *trans*-fatty acids in Australia under alternating assumptions regarding distribution of post-ban *trans*-fatty acid intake.** (DOCX)

## Acknowledgments

The authors acknowledge Ms. Liping Huang for the assessment of potentially iTFA-containing products in the Australian food supply.

## Author Contributions

**Conceptualization:** Matti Marklund, J. Lennert Veerman, Jason H. Y. Wu.

**Data curation:** Miaobing Zheng.

**Formal analysis:** Matti Marklund.

**Investigation:** Matti Marklund, Miaobing Zheng.

**Methodology:** Matti Marklund, J. Lennert Veerman, Jason H. Y. Wu.

**Project administration:** Matti Marklund.

**Resources:** Miaobing Zheng.

**Software:** J. Lennert Veerman.

**Supervision:** J. Lennert Veerman, Jason H. Y. Wu.

**Visualization:** Matti Marklund.

**Writing – original draft:** Matti Marklund.

**Writing – review & editing:** Matti Marklund, Miaobing Zheng, J. Lennert Veerman, Jason H. Y. Wu.

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
