## [Decision Letter · Decision Letter 0]

2 Jun 2020

Dear Dr. Marklund,

Thank you very much for submitting your manuscript "Estimated health benefits, costs, and cost-effectiveness of eliminating industrial trans-fatty acids in Australia" (PMEDICINE-D-19-03970) for consideration at PLOS Medicine. 

[LINK]

In light of these reviews, I am afraid that we will not be able to accept the manuscript for publication in the journal in its current form, but we would like to consider a revised version that addresses the reviewers' and editors' comments. Obviously we cannot make any decision about publication until we have seen the revised manuscript and your response, and we plan to seek re-review by one or more of the reviewers. 

We expect to receive your revised manuscript by Jun 23 2020 11:59PM. Please email us (plosmedicine@plos.org) if you have any questions or concerns.

We look forward to receiving your revised manuscript. 

Sincerely,

Emma Veitch, PhD

PLOS Medicine

On behalf of:

Adya Misra, PhD

Senior Editor 

PLOS Medicine

plosmedicine.org

*We'd suggest revising the title according to PLOS Medicine's style - would recommend adding a subtitle after the study question, summarising the study design (eg :modelling study). 

*We'd recommend noting any key limitations of the study methodology in the abstract, for example stating here that the TFA intake data date back to 2011/2012 and therefore more recent changes in intake may affect interpretability of these findings.

*At this stage, we ask that you include a short, non-technical Author Summary of your research to make findings accessible to a wide audience that includes both scientists and non-scientists. The Author Summary should immediately follow the Abstract in your revised manuscript. This text is subject to editorial change and should be distinct from the scientific abstract. Please see our author guidelines for more information: https://journals.plos.org/plosmedicine/s/revising-your-manuscript#loc-author-summary

*The Academic Editor for your paper commented that it could be made much clearer in the Discussion how the findings add to or contribute additionally beyond current evidence from other modelling studies. Specifically in the Discussion section 'Interpretation and policy implications' it might be an idea to more explicitly start this section with a summary of what current evidence shows and then go on to clarify how the current analyses add evidence beyond that.

*Did your study have a prospective protocol or analysis plan? Please state this (either way) early in the Methods section.

Comments from the reviewers:

Reviewer #1: The authors present their modelling study that attempts to estimate the public health impact and cost-effectiveness of a national ban of iTFAs in Australia, considering both policy costs and reduced IHD burden and health care expenditures. 

There are some typos and grammatical errors to be corrected throughout the manuscript. although overall this is a well written and thorough paper.

The use of a multiple cohort proportional multi-state life table (Markov) model appears to be appropriate and reasonable in this context.

The model formulation provided by the authors looks to be fit for purpose.

The authors have conducted sensitivity analyses, which demonstrate the robustness of the model to many of the key assumptions and parameters. 

Additionally, further strength would be illustrated by including sensitivity analyses surrounding the assumed costs described in the 'Healthcare and policy costs' section. 

"Costs were adjusted to 2010 values, and in the base case analysis, a 3% discount rate was used".

It is noted that a sensitivity analysis of the discount rate from 0-6% has been undertaken by the authors, but can the authors please provide a justification (preferably evidence based) for choosing a discount rate of 3%? 

The authors present their study and findings fairly, including uncertainty clearly in their results, figures and conclusions.

In addition to the statistical analyses completed and presented in the results section, did the authors consider different intervention impacts? Furthermore, what the breakpoint for cost effectiveness is? 

I.e what minimum value does the PIF need to be in order for the intervention to be considered cost effective?

Reviewer #2: Thank you for the opportunity to review PLOS Medicine MS 19-03970, "Estimated health benefits, costs, and cost-effectiveness of eliminating industrial trans- fatty acids in Australia." This paper used a Markov simulation model to estimate the benefits, costs, and cost-effectiveness of a ban on industrial trans-fatty acids in Australia over a 10-year and lifetime time horizon. The authors find that under all scenarios, such a ban would be cost-effective, and under certain assumptions (e.g., 6% discount rate) would be cost-saving. The paper is well-written, well-executed, and speaks to an interesting and important policy question. Main strengths are use of high-quality data sources, evaluation of an important policy question, incorporation of industry costs, and examination of impacts on health disparities. In my read, there are two key limitations, which comprise my major concerns:

Major Concern 1: First, the data used to estimate TFA intake are somewhat old now (2011-2012), and there is no discussion of whether TFA intake may have increased or declined since that time. This limitation could be overcome by either citing evidence that TFA has not changed appreciably since 2011-2012, or by varying estimates of TFA intake in sensitivity analyses (e.g., assume average TFA intake is 25% or 50% or 75% of what it was in 2011-2012 across all groups). 

Major Concern 2: Second, there is no discussion of how companies are likely to reformulate when they remove TFA. If companies increase saturated fat when removing TFA, or add more sodium or sugar, this might offset some of the benefits of lower TFA on IHD incidence and death. This needs to be explored to paint an accurate picture of the effects of TFA on health. 

If these concerns can be addressed, I believe this manuscript would make an important contribute to the literature and would be appropriate for publication in PLOS Medicine. 

My other comments/suggestions are minor.

Abstract:

1. Need to define "iTFA" here.

2. I would suggest you state somewhere in the abstract the policy you're modeling - i.e., a TFA ban. Otherwise, it's not clear what "policy costs" are, since no policy has been mentioned.

Introduction

3. "1% E" - I would spell out energy since you have not defined this abbreviation yet. (I would personally prefer "energy" to "E" throughout, but at very least the abbreviation needs to be defined at first use). 

4. Noted above, but -- are there more recent data on TFA since 2013? It's now almost 7 years later, so it seems possible that there would be additional reductions in TFA in the Australian food supply since 2013, particularly given that other countries have adopted policies since then (so multi-national corporations might reformulate products to, e.g., comply w/ US regulations, even if some of those products are sold in the US).

5. "Our prior modelling suggest at current level of intake, TFA continues to contribute to around 500 deaths per year…" I would clarify this is among Australians. 

6. Would be nice to note whether Australia has considered this policy. (The simulation is useful even if the policy has not been proposed yet, but if it has, would be good to note it's on policymakers' radar). 

7. You might mention here why it's useful to conduct this simulation given Australia's low levels of TFA intake. I really liked this section of the discussion and think including a brief mention of these points in the intro might help motivate the paper, particularly for readers who think we already know what we need to know about TFA bans.

Methods

8. I commend the pre-specification of the subgroups. Is there an associated protocol that describes the pre-specified analyses? If so, would be helpful to include it as a hyperlink or appendix file. 

9. Intervention: It would be good to clarify whether the ban applies to all foods or just packaged foods. Later you notice that the FoodSwitch database does not contain restaurant foods, so I assume the ban would apply to all foods, but good to clarify.

10. Study design: Here or in the introduction, you might give readers a sense of why these subpopulations are meaningful. The Socio-Economic Disadvantage groups are intuitively meaningful, but for non-Australian readers, it's not immediately clear why it matters to examine remoteness.

11. Can you clarify the sample size of the Australian National Nutrition and PA Survey? Was sample size adequate to calculate reliable mean TFA intake for all age and sex groups? 

12. Data sources. In locations where TFA has been banned, is it true that intakes drop to <= 0.5% energy intake? That is, do bans effectively remove all iTFA or do some companies continue to include some TFA in their products? I would state the evidence for this, or vary the assumption in sensitivity analyses. 

13. Data sources: It wasn't clear to me why the intervention would lead to additional annual cost of 1% of the initial reformulation cost - can you clarify?

Results

14. Figure 1. This is a helpful visual. You might consider modifying to use four panels and separating the deaths from the incidences (vs. the current two panels that use different colored bars to show deaths vs. incidences). This might make it easier to see the pattern of results, particularly for SEIFA (where the white bars are small and somewhat overwhelmed by the gray bars). A similar comment would apply to the other tables using this design, but I found Figure 1 most difficult to discern. 

15. Figure 5. This was hard to follow. Consider using color coding to show the different models, instead of the arrows/lines. 

Discussion

16. I appreciate that the study used the most recent data on TFA intake, but these data are relatively outdated now. It would be good to note this as a limitation, particularly if the authors are unable to locate more recent data on trends of TFA content since 2013. 

17. The introduction stated that the authors' previous work has found that TFA contributes to about 500 death/year. The present work suggests banning iTFA would avert about 2,300 IHD deaths over 10 years (about 230/year). I assume the discrepancy is either due to modeling differences or because the 500/year statistic includes non-IHD deaths due to TFA. Consider clarifying/comparing?

[LINK]

---

## [Decision Letter · Decision Letter 1]

9 Sep 2020

Dear Dr. Marklund,

Thank you very much for re-submitting your manuscript "Estimated health benefits, costs, and cost-effectiveness of eliminating industrial trans-fatty acids in Australia: modelling study" (PMEDICINE-D-19-03970R1) for review by PLOS Medicine.

I have discussed the paper with my colleagues and the academic editor and it was also seen again by xxx reviewers. I am pleased to say that provided the remaining editorial and production issues are dealt with we are planning to accept the paper for publication in the journal.

[LINK]

We look forward to receiving the revised manuscript by Sep 16 2020 11:59PM. 

Sincerely,

Barry Popkin, PhD

University of North Carolina 

PLOS Medicine

plosmedicine.org

Requests from Editors:

Title- please can you add “a” to the study descriptor to read “a modelling study”

Throughout- please add punctuation after the reference square brackets please

Throughout the manuscript, whenever you restate (eg at the start of the Discussion) the estimated deaths or cases averted, it should be clear what you are comparing to (the current level of TFA intake).

Rephrase the Abstract conclusions to include "Our model estimates that..." or similar language. Also could include more on the implications, using similarly cautious language. The last point of the Author Summary could perhaps be retooled for the Abstract Conclusions. Please use similar cautious language in the discussion

Page 5- “Generally Recognized as Safe for..” please consider if these words need to be capitalised?

Page 13 “As there have been suggested that…” seems to have been cut off

Please ensure that the study is reported according to the CHEERS guideline, and include the completed checklist as Supporting Information. When completing the checklist, please use section and paragraph numbers, rather than page numbers. Please add the following statement, or similar, to the Methods: "This study is reported as per the xxx guideline (S1 Checklist)." 

Methods- please remove the sub-heading “Intervention” as it is a bit misleading for a modelling study. I recommend paraphrasing the sentence to include the aim of your work including the study design used. 

Methods-please provide further information about the cohorts used in the study. These are very briefly mentioned and it would help to outline the participant demographics in the methods.

Please use table footnotes to add explanations for acronyms

Discussion- I’m not sure what you mean by “population lifetime” could you please clarify in text?

Please start the discussion with a 1-2 sentence summary of what was done before delving into the main findings

I think it is important to distinguish between cost saving and cost effectiveness here- since you aimed to look at cost effectiveness. Please use standard terminology throughout the submission to avoid confusion

Please ensure all acronyms are introduced on first view

Please use Vancouver style for the references and remove all iterations of [Internet] from the reference list 

Data availability statement- please mention the data sources used in the study and provide the code used to undertake analyses in a suitable repository. 

Comments from Reviewers:

Reviewer #1: The authors have responded to each comment in turn, undertaking additional analyses and including the results in the revised manuscript accordingly.

Reviewer #2: Thank you for the opportunity to review the revised manuscript, MS 19-03970, "Estimated health benefits, costs, and cost-effectiveness of eliminating industrial trans- fatty acids in Australia: modelling study." The authors have been exceptionally responsive in their revision. In my initial review, I raised two major concerns: 1) Whether it was reasonable to assume TFA intake has been stable since 2011-2012 (when data underlying this study were collected) and 2) Whether the model needed to account for product reformulation to accurately assess health impacts of a iTFA ban. The authors have adequately addressed these two concerns. First, they provide data to indicate that it is indeed reasonable to assume stable TFA since 2011-2012, and conduct sensitivity analyses showing robustness of results to assuming a lower TFA intake level in the base population. Second, they provide data/references to indicate that RRs for change in TFA intake are not specific to a certain pattern of dietary substitution/replacement -- that is, the contrast used in the RR estimates used in this study do not depend on how companies reformulation when TFA is lowered. They further provided recent reviews of reformulation after reductions/eliminations in TFA, finding no discernible pattern in replacement of TFA by particular types of fat, again suggesting their modeling strategy here was reasonable. 

My other comments in the initial review were more minor, but these too have been adequately addressed. I have no further major comments. I will note that I personally had difficulty discerning some of the colors in Figure 5 (e.g., the pink and red looked nearly identical to me), but I assume this will get sorted by the journal in the next step of the publication process. (Or perhaps a high res version will be sufficient).

[LINK]

---

## [Editor Report · Decision Letter 2]

29 Sep 2020

Dear Dr. Marklund, 

On behalf of my colleagues and the academic editor, Dr. Barry M. Popkin, I am delighted to inform you that your manuscript entitled "Estimated health benefits, costs, and cost-effectiveness of eliminating industrial trans-fatty acids in Australia: a modelling study" (PMEDICINE-D-19-03970R2) has been accepted for publication in PLOS Medicine. 

PRODUCTION PROCESS

Before publication you will see the copyedited word document (within 5 busines days) and a PDF proof shortly after that. The copyeditor will be in touch shortly before sending you the copyedited Word document. We will make some revisions at copyediting stage to conform to our general style, and for clarification. When you receive this version you should check and revise it very carefully, including figures, tables, references, and supporting information, because corrections at the next stage (proofs) will be strictly limited to (1) errors in author names or affiliations, (2) errors of scientific fact that would cause misunderstandings to readers, and (3) printer's (introduced) errors. Please return the copyedited file within 2 business days in order to ensure timely delivery of the PDF proof. 

If you are likely to be away when either this document or the proof is sent, please ensure we have contact information of a second person, as we will need you to respond quickly at each point. Given the disruptions resulting from the ongoing COVID-19 pandemic, there may be delays in the production process. We apologise in advance for any inconvenience caused and will do our best to minimize impact as far as possible.

PRESS

PROFILE INFORMATION

Thank you again for submitting the manuscript to PLOS Medicine. We look forward to publishing it. 

Best wishes, 

Adya Misra, 

Senior Editor 

PLOS Medicine

plosmedicine.org